# Ultrasound-Guided Near-Nerve Needle Sensory Technique for the Diagnosis of Tarsal Tunnel Syndrome

**DOI:** 10.3390/jcm10143065

**Published:** 2021-07-11

**Authors:** Lorena Vega-Zelaya, Álvaro Iborra, Manuel Villanueva, Jesús Pastor, Concepción Noriega

**Affiliations:** 1Avanfi Institute, Calle Donoso Cortes 80, 28015 Madrid, Spain; docalvaroiborra@gmail.com (Á.I.); mvillanuevam@gmail.com (M.V.); 2Clinical Neurophysiology, Hospital Universitario de La Princesa, C/Diego de León 62, 28006 Madrid, Spain; jesus.pastor@salud.madrid.org; 3Unit for Ultrasound-Guided Surgery, Hospital Beata María Ana, 28007 Madrid, Spain; 4Department of Nursery and Physiotherapy, Faculty of Medicine and Health Sciences, University of Alcalá, 28801 Alcalá de Henares, Spain; concha.noriega@uah.es

**Keywords:** tarsal tunnel syndrome, nerve entrapments, near-nerve needle sensory technique, ultrasound-guided, tibial nerve, nerve conduction studies, electrodiagnosis, neurophysiology

## Abstract

Background: Tarsal tunnel syndrome (TTS) is one of the most common entrapment syndromes. Although diagnosis is supported by imaging tests, it has so far been based on clinical findings. Neurophysiological tests are not effective for providing an accurate diagnosis. The objective of this study was to analyze the efficacy of the ultrasound-guided near-nerve needle sensory technique (USG-NNNS) for the diagnosis of TTS Methods: The study population comprised 40 patients referred for a neurophysiological study owing to clinical suspicion of TTS. Routine neurophysiological tests were performed and compared with the results of USG-NNNS. Results: The diagnosis of TTS was achieved in 90% of cases. We found significant differences between lateral plantar sensory recordings with surface electrodes and USG-NNNS techniques for amplitude, nerve conduction velocity (NCV), and duration. As for the medial plantar sensory recordings, differences were found only for duration. No responses were obtained with surface electrode studies in 64.8% of cases. In addition, we observed normal sensory NCV with surface electrodes in 20 patients, although this decreased when the NNNS technique was used. Conclusions: This is the first report of the efficacy of the USG-NNNS technique for confirming the diagnosis of TTS.

## 1. Introduction

Tarsal tunnel syndrome (TTS) is an entrapment syndrome of the entire tibial nerve (TN) or its terminal branches (medial plantar, lateral plantar, and calcaneal nerves) behind the medial malleolus and under the flexor retinaculum (or laciniate ligament) and the deep fascia of the abductor hallucis muscle [1,2]. The tibial nerve is both a motor and a sensory nerve and is the largest distal extension of the sciatic nerve [3].

The symptoms described include a variety of alterations ranging from posteromedial pain and numbness in the sole of the foot, tightness, and cramps that are initially intermittent but can be accentuated by prolonged standing or walking [4]. Pain at night is common and often severe enough to awaken the patient.

Clinical signs on physical examination include posteromedial tenderness over the nerve, a positive Tinel sign, and, in some cases, bulging of the retinaculum. Objective negative signs, including hypoesthesia and claw toes, may also be observed [1,4,5].

Although TTS is the fifth most published entrapment syndrome [6], its etiology, prevalence, and diagnosis are controversial [6,7].

Diagnosis is currently by physical examination supported by another test. Diagnostic imaging such as computed tomography (CT) may reveal a lesion, although magnetic resonance imaging (MRI) could prove to be a more helpful technique for evaluating the peripheral nerves inside the tarsal tunnel [8,9]. High-resolution ultrasound is almost 100% effective as a tool for identifying the tibial nerve and its branches (Figure 1), as the authors have previously published [10,11]. It is also inexpensive, accessible, and quick, although it has proven to be particularly useful for the treatment of TTS [10,12,13].

Neurophysiological tests are more complex and uncertain, since traditional examinations have poor diagnostic sensitivity for this disease [6]. Despite the availability of recommendations from the American Association of Neuromuscular and Electrodiagnostic Medicine (AANEM, formerly AAEM) on the neurophysiological techniques recommended to confirm the presence of TTS [5], this approach is not as reliable as in other nerve entrapment syndromes [1,13].

The near-nerve needle sensory technique (NNNS), which was first described more than 50 years ago [14], has been used for the diagnosis of several conditions, including TTS [15,16,17,18]. It has the advantage that it makes it possible to examine the most distal sensory nerves of the foot and may identify neuropathies in the early stages [19]. The principal disadvantage is that it is invasive, painful, and time-consuming. The use of ultrasound to properly identify the recording site could reduce the duration of the test and make the study less painful for the patient.

The objective of this study was to analyze the electrophysiological results of patients with symptoms suggestive of TTS by comparing the neurophysiological techniques recommended by the AANEM for the diagnosis of TTS and the ultrasound-guided NNNS technique (USG-NNNS).

## 2. Materials and Methods

### 2.1. Patients

We performed a prospective study over a period of 10 months. The study population comprised 40 consecutive patients (19 males and 21 females) with a mean age of 44.5 ± 2.0 years (range 19–73), who were referred for a neurophysiological study owing to clinical suspicion of TTS. A detailed clinical history was taken and a physical examination carried out by physicians and specialists from a referral center in orthopedic surgery and podiatry with extensive experience in the diagnosis and treatment of complex diseases. All the patients had undergone MRI and ultrasound to rule out other conditions and space-occupying lesions that could have compromised the nerve.

As all the neurophysiological studies were bilateral, we obtained a total of 80 recordings of the TN. The analysis excluded symptomatic patients with normal values in neurophysiological studies. We compared sensory nerve conduction velocity (NCV) results in USG-NNNS with those of surface electrodes in the symptomatic group. The findings for the other techniques were also reported. As the symptoms were unilateral in some patients, we used the records of the asymptomatic sides and compared the findings with those from the symptomatic group.

### 2.2. Electrodiagnostic Techniques

The neurophysiological study (Tru Trace^®^ EMG, Deymed Diagnostic, Kudrnáčova, Czech Republic) was conducted using a combination of tibial motor, sensory, and mixed medial and lateral plantar NCV with surface electrodes. We also performed tibial sensory medial and lateral plantar NCV with USG-NNNS. Other neurophysiological techniques were used to rule out lumbar radiculopathy and peripheral neuropathy. The reference values used have been published elsewhere [20,21].

#### 2.2.1. Tibial Motor Nerve Conduction Velocity (NCV)

The compound muscle action potential was recorded over the abductor hallucis and abductor digiti minimi pedis muscles using surface electrodes (bandwidth 50–3000 Hz and notch on) and elicited by electrical stimulation (constant-current) of the TN at the medial ankle. The pulse duration was 200 μs with an intensity of 20–60 mA. We measured distal onset latency, amplitude, and motor conduction velocity.

#### 2.2.2. Medial and Lateral Plantar Sensory NCV with Surface Electrode

The medial and lateral plantar sensory nerve action potential (SNAP) was recorded orthodromically at the medial malleolus with SEs (surface electrodes, bandwidth 10–1500 Hz, notch on). The first and fifth digital nerves were stimulated (constant-current) with ring electrodes at 4.1 Hz and a pulse-width of 100 μs. Supramaximal stimulation was ensured by increasing the current 25–30% beyond the maximal stimulation intensity when the SNAP was observed on each stimulus. At least 100–200 stimuli were averaged in each recording in at least 3 consecutive runs. Onset latency was measured from the stimulus to the initial negative deflection from baseline for biphasic SNAPs or to the initial first positive peak for triphasic SNAPs. For maximum NCV (MaxNCV), we considered onset latency (the main positive peak). Furthermore, amplitude was measured from peak to peak, and duration from onset of the first positive peak to the baseline return of the last component of the potentials.

#### 2.2.3. Medial and Lateral Plantar Mixed NCV

We obtained a mixed response in the orthodromic recording at the medial malleolus and by stimulating the medial sole (medial plantar nerve) and lateral sole (lateral plantar nerve) at 14 cm from the recording electrodes. Settings were the same as for the SNAP recordings. We measured peak latency, amplitude, and conduction velocity.

#### 2.2.4. Ultrasound-Guided Near-Nerve Needle Sensory (USG-NNNS) Technique

The instrument set included an ultrasound device (Alpinion ECube15) with an 8–17–MHz linear transducer and the Needle Vision Plus™ software package (Alpinion Medical Systems, Bothell, WA, USA). The orthodromic recording of the SNAP was based on signal averaging techniques and the settings described above. The active needle recording electrode was inserted and placed under ultrasound guidance close to the tibial nerve in the ankle above the flexor retinaculum. The reference needle electrode was placed subcutaneously at the same level as the active electrode at a transverse distance 3–4 cm (Figure 2). The adequacy of the needle position was confirmed by stimulating the nerve through the active electrode. The correct position was determined when the big toe was contracting minimally with less than 1 mA for a stimulus of 50 μs in duration. Then, digits I and V were stimulated separately. We used the same settings and the same measurement variables for surface registration. We also measured the minimum NCV (MinNCV) taking into consideration the last positive peak of the potential (Figure 3).

### 2.3. Statistical Analysis

Statistical comparisons between groups were performed using the *t* test (two groups) or analysis of variance (more than two groups) for parametric samples. For non-parametric samples, the Mann–Whitney test (two groups) and the Kruskal–Wallis test (more than two groups) were used. Normally distributed values were assessed using the chi-square or Kolmogorov–Smirnov test.

SigmaStat 3.5 software (SigmaStat, Point Richmond, CA, USA) was used for statistical analysis. Significance was set at *p* < 0.05. The results are presented as the mean ± standard error of the mean (SEM), except where otherwise indicated.

## 3. Results

### 3.1. Clinical Features

Symptoms were bilateral in 25 patients and unilateral in 15. The electrodiagnostic techniques yielded normal findings in four patients, of whom two had bilateral symptoms and were not included in the data analysis. Regarding the other two patients, we included the records of the asymptomatic side for the normal group. Thus, TTS was diagnosed in 90% of cases. In total, we analyzed 74 records from TN.

Patients complained of pain in the area of the ankle, heel, and sole, as well as numbness, radiating pain, and paresthesia along the distribution of the tibial nerve and its branches. Symptoms usually worsened at night, when walking, after prolonged standing, or after physical activity and generally improved after rest, with variations depending on the sites affected and severity of the nerve compression.

The Hoffman–Tinel test was positive in more than half of patients with the condition. In addition, the Valleix phenomenon (constant pressure on the nerve) can lead to radiating pain with tingling and numbness in about one third of cases.

The MRI and ultrasound studies did not reveal enlargement of the nerve or other anatomical abnormalities of the nerve or its branches.

### 3.2. Electrophysiological Findings

USG-NNNS revealed pathological findings in 59 recordings. The combination of medial and lateral plantar abnormalities was the most common finding, although only the medial and plantar nerves were affected in 12.9% and 8.5%, respectively.

Regarding the neurophysiological test with surface electrodes, no responses were obtained from either the medial or lateral nerve in 27 recordings (47.8%). In addition, isolated absence of response of the lateral plantar branch was observed in 10 patients (17%).

#### 3.2.1. Comparison between the Surface Electrode and USG-NNNS Techniques

The mean amplitude of the lateral nerve was greater with USG-NNNS (1.0 µV ± 0.11; range: 4.0–0.1 µV) than with surface electrode (0.65 µV ± 0.09; range 2–0.2 µV, *p* < 0.05), and the mean sensory NCV obtained from the plantar lateral nerve was significantly lower with USG-NNNS (32.1 m/s ± 0.9 vs. 36.7 m/s ± 1.3; *p* < 0.001). In addition, the duration of SNAP for the medial and lateral plantar nerve was significantly longer with USG-NNNS (4.9 ± 0.2 ms vs. 2.0 ± 0.06 ms and 5.4 ± 0.2 ms vs.1.90 ± 0.09 ms, respectively, *p* < 0.001; Figure 4). No differences were found in terms of amplitude and sensory NCV (SNCV) for the medial plantar branch (Table 1).

The average intensity at which we confirmed the correct position of the active needle electrode, i.e., when digit I was contracting minimally, was 0.6 ± 0.03 mA (range: 0.2–1 mA). In all cases, a single puncture was necessary to reach the correct site, although on some occasions the needle was repositioned (without the need for a new puncture).

A key finding was the normal sensory NCV with surface electrodes in 8 patients for the medial nerve and in 12 for the lateral nerve, although with the NNNS technique, SNCV was decreased.

Isolated sensory alteration of the TPN was the most frequent type (69.5%). The remaining cases (30.5%) comprised a combination of sensory and motor abnormalities. The findings for the other neurophysiological tests are shown in Table 2.

#### 3.2.2. Comparison between the Symptomatic Group and the Asymptomatic Groups

Regarding the results of the asymptomatic side group (17 recordings) for the study of sensory conduction of the TN, no responses were obtained with surface electrodes in 12 cases (70.5%): in 11 patients this applied to both the medial and the lateral nerve and in one case the absence of response was only for the plantar lateral nerve. Furthermore, in four patients in whom responses were obtained, the amplitude was below the limit of normality for the medial and lateral plantar nerves [21,22]. The mean amplitude for this group was 1.32 ± 0.3 µV for the medial nerve and 0.5 ± 0.2 µV for the lateral nerve.

Regarding the results of the NNNS technique in this group for the medial nerve, we found a mean amplitude of 1.51 ± 0.3 µV, the MaxNCV was 39.8 ± 0.7 m/s, and the MinNCV was 30.58 ± 0.8 m/s. The mean duration was 2.57 ± 0.2 ms. As for the lateral nerve, the mean amplitude was 0.8 ± 0.3 µV. The MaxNCV and MinNCV were 41.1 ± 0.7 m/s and 29.0 ± 1.0 m/s, respectively, with a duration of 3.9 ± 0.3 ms.

When we compared these results with those of the group of symptomatic patients, we found significant differences with respect to the MaxNCV, MinNCV, and duration for both the medial and the lateral nerve (*p* < 0.001). Differences in amplitude were not significant (Figure 5).

## 4. Discussion

We show the usefulness of USG-NNNS for the diagnosis of distal neuropathy of the TN. We were able to make a positive diagnosis in 90% of patients. The effectiveness of the NNNS technique in diagnosis of TTS has been discussed elsewhere [15]. Despite being a sufficiently useful technique for diagnosing several conditions [15,16,18], application of USG-NNNS is not standardized, since it is a time-consuming and extremely painful method. The novelty of our study is that the nerve is localized using ultrasound, which makes it faster, more efficient, and less uncomfortable for the patient than the NNNS technique.

Involvement of both branches of the TN is the most common finding, although isolated alterations can also be found.

One of the most interesting findings in terms of comparing surface electrodes and USG-NNNS is that in almost 65% of cases, non-reproducible potentials were obtained with the surface electrodes. Up to 17% were not acquired in isolation in the lateral plantar nerve, thus clearly reducing the reliability of the study with SEs to make a positive or negative diagnosis.

In addition, when we analyze both techniques, significant differences are found for amplitude, SNCV, and duration in the lateral plantar nerve. Therefore, the potentials obtained with the NNNS technique are of greater amplitude, lower SNCV, and longer duration. This is explained by the fact that with surface electrodes, we found SNCV parameters to be normal in 12 patients for the lateral nerve. In other words, neurophysiological tests with SEs can yield false-negative results in a non-negligible percentage.

Moreover, in eight patients, normal SNCV parameters were found with surface electrodes for the medial nerve compared with USG-NNNS. However, the only significant differences were for the duration of the potential (Table 1).

Another remarkable finding is that, in asymptomatic patients, no reproducible responses were obtained with surface electrodes in more than 70% of cases, thus potentially confirming the lack of efficacy of this neurophysiological test, which also seems to yield false-positive results. This finding was also reported for symptomatic patients.

Regarding the comparison of the two techniques between the asymptomatic and symptomatic groups, we found significant differences for Max NVC, MinNCV, and duration. It is not surprising that the difference in amplitude between the two techniques was not significant, since the amplitude of the potential with the NNNS depends on the distance and position of the needle with respect to the nerve and on the conduction distance (Appendix A).

There is no question that the NNNS technique provides invaluable information that cannot be obtained with surface electrodes. Its main advantages are a lower signal-to-noise ratio, the ability to register potentials of 1 microvolt, and even a MaxNCV of 20 m/s.

We found that maximum and minimum conduction velocities can be confidently recorded in patients with TTS. Nonetheless, it is extremely important to ensure that the active electrode is placed near the nerve. We addressed this goal using a USG-based approach. As our numerical simulations showed, even without true physiological values (which are not necessary to demonstrate our hypothesis), a change in dipole length is sufficient to obtain a smaller action potential. From Equation (A1) (Appendix A), it is evident that amplitude decreases with the inverse of the square of distance [23,24]. Therefore, small potentials will be more sensitive to distance than higher ones, as can be observed from the figure in the Appendix. Other important effects for amplitude include synchronization, nerve diameter, and membrane current density. However, the most relevant is dipole dependence on distance [25].

The two major disadvantages of the NNNS technique—extreme pain on insertion of the needle and longer duration—mean that it is rarely used. By introducing ultrasound to this technique, both disadvantages are considerably diminished, since in most cases the needle is only inserted once, and positioning takes less than 1 min. Consequently, the advantages of USG-NNNS clearly outweigh the few disadvantages and make for a more viable and consistent neurophysiological technique that achieves a more reliable diagnosis and follow-up of TTS.

This information is of extraordinary value, since nerve conduction studies with the NNNS technique are not routinely used to rule out or confirm TTS. Therefore, in most cases, the clinician must decide on a diagnosis and treatment based on clinical findings. The introduction of ultrasound to locate the appropriate site of the active electrode enables a much more viable and consistent neurophysiological technique that ensures a more reliable diagnosis and follow-up.

Importantly, no gold standard test is available, and although there are recommendations on the neurophysiological tests that must be used for confirming the TTS [5], the poor diagnostic sensitivity of the instrumental examinations have led clinicians to make a diagnosis based mainly on the medical history and physical examination. Therefore, many authors believe that this condition is underdiagnosed [26,27].

The technique we describe here could completely change our approach to the diagnosis and follow-up of TTS.

It should be mentioned that the main limitation of our study is that it does not have a real control group of asymptomatic patients for a better comparison. This situation would have been the ideal, although our real objective was to describe the diagnostic capacity of the USG-guided near-nerve technique.

Our vision of neurophysiological studies, as performed today, is one based on wide-ranging changes in which the technique is converted and adapted for other conditions and for exploring other, potentially more accessible nerves, especially those that affect sensory fibers early. To the best of our knowledge, the USG-NNNS technique has not been used to localize other nerves. Ours is the first description of this technique.

## 5. Conclusions

Ours is the first report on USG-NNNS being used to confirm the diagnosis of TTS. According to our results, diagnosis should be based on MaxNVC, MinNVC, and temporal dispersion. We were able to diagnose TTS in 90% of patients. In addition to being more reliable, this technique has proven to be less painful and faster to perform than the classic NNNS technique. Our results are reproducible for the study and diagnosis of other neuropathies.

## Figures and Tables

**Figure 1 jcm-10-03065-f001:**
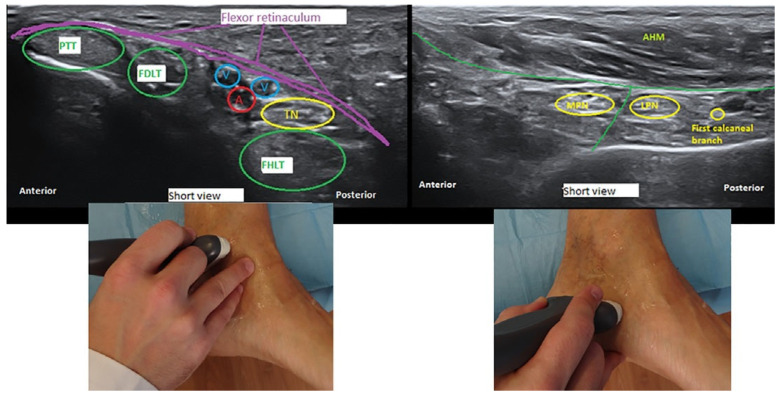
Ultrasound delineation of the anatomy of the tarsal tunnel: transverse or short-axis view (cross section) at the proximal and distal tarsal tunnel. A indicates posterior tibial artery; FDLT, flexor digitorum longus tendon; FHLT, flexor hallucis longus ten don; PTT, posterior tibial tendon; AHM, abductor hallucis muscle; V, posterior tibial vein; TN, tibial nerve; MPN, medial tibial nerve; and LPN, lateral plantar nerve.

**Figure 2 jcm-10-03065-f002:**
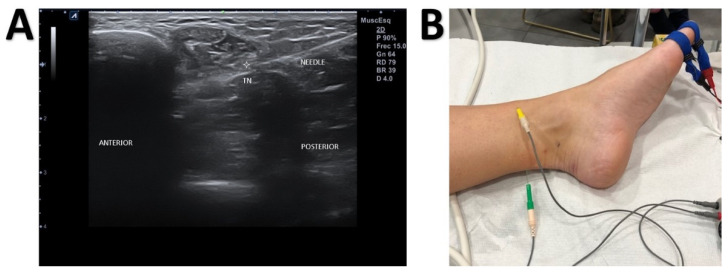
Ultrasound-Guided Near-Nerve Needle Sensory (USG-NNNS) Technique. (**A**) Ultrasound image showing the position of the needle relative to the tibial nerve (TN). (**B**) Placement of the active recording needle (green) and reference electrodes (yellow) for the near-nerve needle sensory (NNNS) technique.

**Figure 3 jcm-10-03065-f003:**
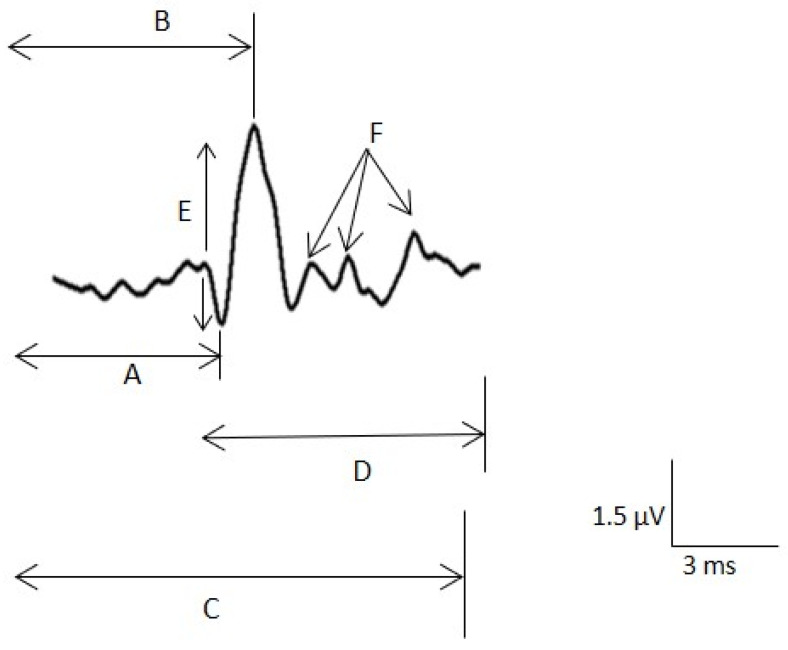
Sensory nerve action potential (SNAP) measurements. A. Latency to the main positive peak for the maximal nerve conduction velocity (NCV). B. Latency to the negative peak. C. Latency to the last negative for the minimal NCV. D. Duration of the potential. E. Peak-to-peak amplitude. F. Small components.

**Figure 4 jcm-10-03065-f004:**
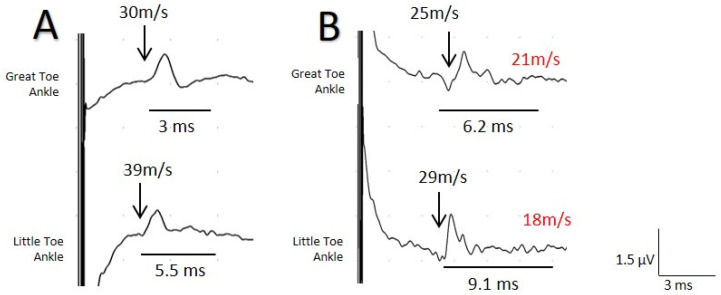
Sensory nerve action potentials (SNAP) recorded for the same patient with surface electrodes (**A**) and the near-nerve needle technique (**B**) at the level of the medial malleolus with stimulus at the I and V digits separately for the evaluation of the medial and lateral plantar branch. A clear difference is observed between the two tests for the maximum NCV (MaxNCV) and duration, although the greater difference is more evident for the lateral plantar branch. minimum NCV (MinNCV) is shown in red.

**Figure 5 jcm-10-03065-f005:**
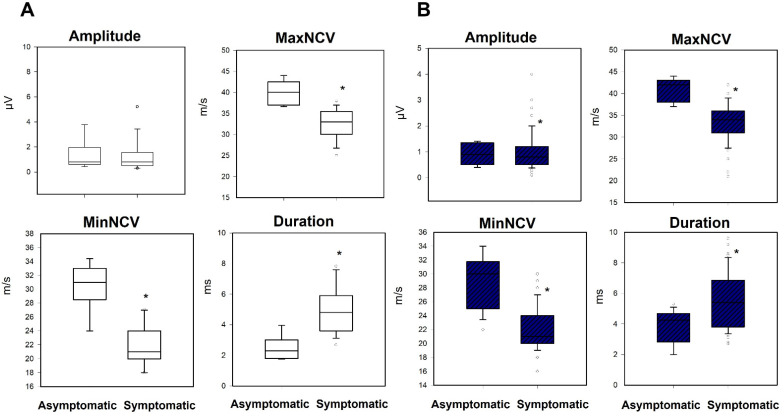
Box-plots comparing the asymptomatic and symptomatic groups for the variables amplitude, maximum, minimal conduction velocity and duration, for the (**A**) medial and (**B**) lateral nerve. Black asterisk = difference between asymptomatic/symptomatic.

**Table 1 jcm-10-03065-t001:** Comparison between SE and ultrasound-guided (UGS)-NNNS NCV techniques.

	SE NCV		NNNS NCV		*p* Value	
Variables	Medial	Lateral	Medial	Lateral	Medial SE/NNNS	Lateral SE/NNNS
Amplitude (µV)	0.98 ± 0.15	0.65 ± 0.09	1.40 ± 0.22	1.00 ± 0.11	*p* = 0.191 *	*p* = 0.031 **
SNCV (m/s)	33.91 ± 0.69	36.73 ± 1.29	32.51 ± 0.51	32.14 ± 0.99	*p* = 0.105 *	*p* = 0.012 **
Duration (ms)	2.04 ± 0.06	1.90 ± 0.09	4.3 ± 0.1	5.48 ± 0.29	*p* = <0.001 **	*p* = <0.001 **
No response	27	37 ^£^	2	4 ^¥^		

NCV: nerve conduction study; NNNS: near-nerve needle sensory; SE: surface electrode; SNCV: sensory nerve conduction velocity. ^£^ In 10 cases the absence of response was isolated in the lateral nerve. ^¥^ In 2 cases the absence of response was only for the lateral nerve. * *t* test. ** Mann–Whitney test.

**Table 2 jcm-10-03065-t002:** Results of the complementary neurophysiological tests.

	MNC		Mixed NCV	
Variables	Medial	Lateral	Medial	Lateral
Latency (ms)	4.9 ± 0.1	5.3 ± 0.1	3.6 ± 0.1	3.9 ± 0.1
Amplitude (µV)	7.6 ±0.4	4.9 ± 0.3	3.0 ± 0.5	2.0 ± 0.1
NCV (m/s)	47.5 ± 0.8	46.6 ± 0.8	41.3 ± 0.1	41.7 ± 1.3
Duration (ms)	5.3 ± 0.1	5.11 ± 0.2	1.6 ± 0.02	1.7 ± 0.04
No response (*n*)	0	0	7	10 ^€^

MNC: motor nerve conduction; NCV: nerve conduction velocity. ^€^ No response in three cases isolated.

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
