# Peer review of "Ultrasound-Guided Near-Nerve Needle Sensory Technique for the Diagnosis of Tarsal Tunnel Syndrome"

_jcm, 2021, doi:10.3390/jcm10143065_

Round 1
Reviewer 1 Report
In this study, Vega-Zelaya et al. examined 40 patients referred with the suspicion of tarsal-tunnel syndrome. Patients were examined extensively including ultrasound guided near-nerve recordings. All patients were examined bilaterally.
This is a nice study with clinical relevance. I have the below comments.
Abstract: ”No responses were obtained with surface electrode studies in almost 65% of cases.” A range could be better rather than almost 65%.
Introduction: No comment
Material and Methods: The authors have excluded from the analysis symptomatic patients with normal values in neurophysiological studies. It is not clear why this has been done.
The recordings from the asymptomatic sides were compared with the recordings from the symptomatic legs. Although this is interesting to know, the results have to be compared with a control group. The main limitation of this study is the lack of a healthy control group.
For near nerve recordings, both the maximum and minimum CV were measured. In my experience, it is very difficult to distinguish the last positive peak and measure a minimum CV even if you average more than 200-300 times.
Results: The electrophysiological results could be presented under sub-titles. It is sometimes difficult to follow. The authors should mention the symptoms and results of the neurological examination.
It has been indicated that all the patients had undergone MRI and ultrasound. It could be interesting to know whether there were any enlargement of the nerve, etc?
Discussion: A paragraph on limitations of the study should be added. Among limitations, lack of helathy control group should be mentioned.
Reviewer 2 Report
Dear authors,
many thanks for the possibility to review your work entitled "Ultrasound-guided near-nerve needle sensory technique for the diagnosis of tarsal tunnel syndrome", which deals with the important topic of the TTS.
General comments: Overall the topic of the work is nice, especially the idea of using an USG-NNNS approach for diagnosing an TTS. The MS is quite well written and has a good structure, a minor spell check should be done. the scientific method used has a good originality and novelty. Unfortunately some concerns and open questions remain:
Specific comments: - first of all and important: which anatomical structure is the "posterior tibial nerve" (PTN), throughout the hole MS? Is there any "anterior tibial nerve"? As far as I know, no. So, a thorough anatomical reorganization is mandatory for anatomical clarification using dissection-figures, US figures etc etc., which also would be nice in this study. There are already some good anatomical papers in literature concerning the TT, which should be integrated and cited in the Intro also:
Anatomic Delineation of Tarsal Tunnel Innervation via Ultrasonography
Clinical-anatomic mapping of the tarsal tunnel with regard to Baxter's neuropathy in recalcitrant heel pain syndrome: part I.
Figures and tables OK
Once again thank you for this submission and the possibility of reviewing this work.
Author Response
"Please see the attachment.

Round 2
Reviewer 1 Report
All my comments have been replied. I do not have further suggestions.
Author Response
We appreciate all the reviewer's comments
Reviewer 2 Report
Dear authors,
many thanks for your revision and addressing to the reviewers comments. I have a minor but important comment: again, there is NO posterior tibial nerve (PTN), even if in some scientific papers named like this; this is - also anatomically seen - wrong and should be corrected throughout the hole paper. There is an official "anatomical terminology" which all scientific works should be complied to, being a base for medical communication! Inconsistencies and discrepancies in regard especially to "clinical terminology", and a self-created terminology should be avoided! Therefore, a scientific essential principle should be considered: learn and communicate the international anatomical terminology! "Terminología Anatomica" is a list of names compiled for scientists worldwide that represents a common language for referring to structures of the whole body. Unfortunately, some authors and reviewers ignore this, often unconsciously. This becomes more complicated when different fields or even specialists within one field use different interpretations. Therefore, the official terminology should be respected.
Thank you very much once again for this nice study!
Author Response
We are very grateful to the reviewer for the comment and reflection.
We have changed the term to the correct one, suggested by the reviewer